# Compress and Mix: Advancing Efficient Taxonomy Completion with Large Language Models

## Abstract

Taxonomy completion aims to integrate new concepts into existing taxonomies by determining their appropriate hypernym and hyponym. While semantic and structural information are crucial for this task, existing approaches often struggle to balance these aspects effectively. In this paper, we propose **COMI**, an efficient taxonomy completion framework that leverages large language models (LLMs) to capture both semantic and structural information in a unified manner. COMI **co**mpresses node semantics into token representations, enabling LLMs to efficiently process the input structure composed of these tokens. To enhance the model's understanding of the structure, a further fine-tuning process using contrastive learning with **mi**xup data augmentation is applied, where mixup generates diverse and challenging negative samples. Through these innovations, COMI improves the integration of semantic and structural information, leading to more accurate taxonomy completion. The experimental results on three real-world datasets demonstrate that COMI achieves state-of-the-art performance while showing up to 284× faster inference compared to the previous best method. Our code and compressed tokens will be available for further study upon publication.

## CCS Concepts

• **Computing methodologies → Information extraction**.

## Keywords

Taxonomy Completion, LLM, Context Compression, Mixup

## 1 Introduction

A taxonomy is a tree-like hierarchical structure organized around hypernym-hyponym ("*is-a*") relations between concepts. It has become increasingly popular in many web services because it is widely regarded as capable of indexing and structuring knowledge. Many applications could be found in various downstream tasks, such as product search [61] and recommendation [72], web content tagging [21, 33] and web searching [62]. For example, web search engines use taxonomies to improve search quality and content categorization [20, 62]. Maintaining taxonomies manually by domain experts is labour-intensive and time-consuming, especially as new concepts continuously emerge. To address this, significant research has focused on the *taxonomy completion (TC)* task [1, 12, 52, 70], where

*Conference WWW'25, 03 April–05 May, 2025, Sydney, NSW*
© 2024 ACM.
ACM ISBN 978-1-4503-XXXX-X/25/04

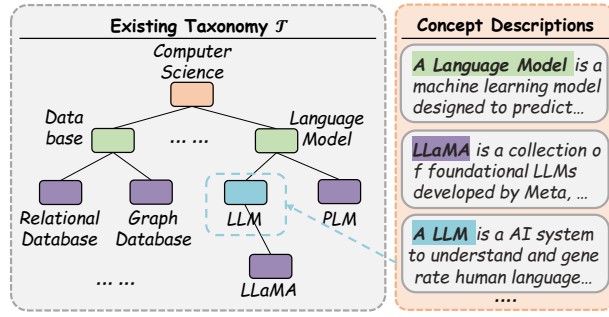

**Figure 1: An example of completing the new concept "LLM" to the existing "Computer Science" taxonomy.**

new concepts (*queries*) are inserted to the most suitable *position* in the existing taxonomy, which composes of a pair of hypernym (*parent*) and hyponym (*child*). As illustrated in Figure 1, for the *query* concept "LLM", it is inserted between the *parent* "Language Model" and *child* "LLaMA" based on the semantic hierarchy.

In taxonomy completion, researchers typically approach the task from two perspectives: semantic and structural. Semantically, hypernyms represent broader concepts, while hyponyms are more specific, with concepts at the same level sharing similar granularity. Structurally, taxonomies follow a tree-like organization where nodes along the path from root to leaf follow a strict hypernym-hyponym order, reflecting increasingly fine-grained abstraction. Thus, the semantic aspect captures differences in meaning between nodes, while the structural aspect reflects their topological relationships. Methods such as CoSTC [31] and TacoPrompt [57] leverage pre-trained language models (PLMs) to capture hypernym-hyponym semantics using concept descriptions, showing strong performance. In contrast, approaches like TaxoEnrich [12] and TEMP [24] model structural information by concatenating node names in path sequences, with the latter achieving better results benefiting from PLMs. However, these methods lack semantically rich descriptions, limiting their effectiveness. Graph-based methods, TaxoExpan [39] model substructures using local Egonet show promise but struggle to align semantic and structural spaces. The key challenge in TC, therefore, lies in effectively integrating both semantic and structural information.

Recently, large language models (LLMs) have demonstrated impressive abilities in semantic understanding and sequence modeling [23, 73], showing great potential for extracting richer semantic and structural information for this task. However, one major challenge with using LLMs is that their input consists of text sequences, making it difficult to directly model graph-structured data. While we can sample path sequences to represent taxonomic structures, simply concatenating the definitions of nodes is not an optimal approach. This straightforward string concatenation introduces two

key problems: the input becomes too long, slowing down inference, and the loss of structural clarity introduces noise. Therefore, this paper investigates *how to efficiently and elegantly leverage the power of LLMs to integrate both semantic and structural information.*

In this paper, we propose COMI, an efficient taxonomy completion framework leveraging LLM's remarkable capabilities. Our approach first compresses semantic information to enable LLMs to handle longer sequences with reduced memory and latency costs. Specifically, we represent each taxonomy node with a single word token, allowing the LLM to compose path sequences from these tokens, capturing structural information while preserving node boundaries and hierarchical relationships. To ensure semantic space consistency, compressed word tokens are generated directly using the LLM based on their descriptions. To facilitate the model's understanding of these compressed tokens, we apply task-specific query-position semantic alignment, which compresses surrounding node information into each token, further enriching their expressiveness. After compression, to enhance the model's understanding of path sequences, we further fine-tune the model using contrastive learning combined with mixup data augmentation. Contrastive learning, effective in discriminative tasks, assists the model capture similarities between instances [53]. To fully leverage this, we introduce a mixup augmentation strategy to generate diverse and challenging negative samples for fine-grained path sequence discrimination. This process includes cut-based input-level mixup, which replaces subsequences in input path sequences, and manifold-level linear mixup, which blends sample representations in the feature space. Through this, the model can better capture fine-grained relationships in path sequences and improve taxonomy completion performance.

We highlight our contributions as follows:

- We propose **COMI**, a novel and efficient framework that leverages the **LLM** to jointly capture both semantic and structural information for taxonomy completion, addressing these two aspects in a unified and integrated manner.
- Our framework achieves efficient LLM inference and flexible path sequence composition through **semantic compression**. To enhance the model's ability to understand path sequences, we introduce two **mixup data augmentation** strategies that help capture fine-grained relationships in path sequences during contrastive learning.
- Experimental results on three real-world datasets demonstrate the superiority of CMOI in both **effectiveness** and **efficiency**. COMI consistently achieves state-of-the-art performance while showing up to 284× faster inference compared to the previous best method.

## 2 Related Work

**Taxonomy Expansion and Completion.** To reduce the computational and expert costs of building taxonomies from scratch, [39] introduced the *taxonomy expansion* (*TE*) task, which focuses on placing emerging concepts as leaf nodes under the most suitable parent in existing taxonomies. This task has gained significant attention and progress [5, 13, 24, 25, 27–29, 35, 36, 40, 42, 44, 51, 55, 56, 64, 66, 68, 74]. To address more practical needs, [70] proposed the *taxonomy completion* (*TC*) task, which inserts emerging concepts as intermediate nodes, linking them between parent and child nodes

in a taxonomy. Further work has explored TC task variants, such as ATTEMPT [54] first identifies a parent and then locates its children, and GenTaxo [67] and ICON [41] generate new concepts based on existing taxonomies.

Taxonomy completion research typically follows two approaches: *Interaction-based* and *Representation-based*. Interaction-based methods, such as TEMP [24], which integrates paths from the root to the candidate parent, and TacoPrompt [56], which uses triplet semantic matching with descriptions of parent, child, and query, are effective but computationally expensive. Representation-based methods, which independently encode the query and candidate position, are more efficient and have become mainstream [1, 31, 70]. For example, QEN [52] generates concept descriptions using PLMs, and TaxoEnrich [12] incorporates ancestral and descendant paths for contextualized representations. TAXBOX [60] employs geometric scorers in box embeddings. However, these models often underperform compared to interaction-based approaches [57]. In this paper, we leverage the semantic knowledge and sequence modeling capabilities of LLMs for representation-based taxonomy completion, achieving results better than interaction-based methods.

**Context Compression in LLMs.** Context compression techniques in LLMs aim to condense explicit inputs into implicit vectors, allowing the model to use these compressed representations efficiently. One line of work focuses on enhancing LLM efficiency by compressing (i) task instruction prompts [6, 18, 30] and (ii) task-relevant inputs [8]. The former enables prompt reuse across various inputs, while the latter retains essential task information for use across multiple prompts. Both approaches reduce input length, improving latency and GPU memory usage during inference. Another approach maps non-text inputs into the LLM's representation space, leveraging its knowledge, reasoning, and sequence modeling capabilities [23, 34, 37, 45]. For example, GraphToken [34] compresses graph structures into tokens for graph reasoning, and AutoTimes [23] converts time series data into token sequences for autoregressive prediction. In this paper, we introduce a task-specific semantic compression method that efficiently integrates structural and semantic information for LLM-based taxonomy completion, enabling more effective path sequence modeling.

**Mixup Augmentation.** Mixup [69] has proven to be an effective data augmentation technique across various domains and tasks [11, 15, 32, 59, 63] for robust representation learning. It generates virtual samples by performing a simple convex combination of data pairs. Based on the level of feature mixing, existing techniques can be broadly categorized into two groups [4]: (i) *global* methods, such as Mixup [69], which mix entire training examples; and (ii) *local* approaches, such as Cutmix [65], which focus on partial feature-level combinations. Mixup [69] combines input data and their labels through convex interpolation, while Manifold Mixup [49] extends this to hidden representations. Cutmix [65] replaces a region of one image with a patch from another, adjusting their labels proportionally to the mixed area. Global mixing approaches [17, 32, 48, 71] encourage the model to learn holistic patterns, whereas local techniques like Cutmix and its variants [22, 50, 63] enhance the model's ability to capture fine-grained, localized features. From these observations, we propose a mixed sample data augmentation method that naturally combines Mixup and CutMix, so that it can take advantage of both methods for fine-grained structure discrimination.

# 3 Methodology

In this section, we formalize the taxonomy completion task (§3.1) and present the proposed **COMI** framework. As illustrated in Figure 2, the framework comprises two stages: first, we conduct taxonomic semantic compression to enable efficient LLM use and flexible path composition (§3.2); second, we fine-tune the model using contrastive learning (§3.3), incorporating a mixup data augmentation strategy (§3.4) to enhance path discrimination.

## 3.1 Problem Formulation

*Definition 3.1 (**Taxonomy**).* A taxonomy $\mathcal{T} = (\mathcal{N}, \mathcal{E})$ is a directed acyclic graph (DAG), $\mathcal{N}$ and $\mathcal{E}$ denote its set of nodes and edges, respectively. Each node $n \in \mathcal{N}$ represents a unique concept, defined based on a supporting corpus $\mathcal{D}$. A directed edge $\langle n_p, n_c \rangle \in \mathcal{E}$ indicates a hypernym-hyponym relationship, where the parent node $n_p$ corresponds to a more general concept, and the child node $n_c$ represents a more specific concept.

*Definition 3.2 (**Taxonomy Completion**).* Suppose that we have an existing taxonomy $\mathcal{T}^0 = (\mathcal{N}^0, \mathcal{E}^0)$ which comprises nodes $\mathcal{N}^0$ and edges $\mathcal{E}^0$. Given a set of new concepts $C$ and a comprehensive corpus $\mathcal{D}$ that defines both the existing nodes $n \in \mathcal{N}^0$ and the new concepts in $C$, the objective is to extend $\mathcal{T}^0$ into a completed taxonomy $\mathcal{T}$. This is achieved by revising the structure, i.e., removing outdated edges and introducing new ones to appropriately integrate the new concepts, resulting in $\mathcal{T} = (\mathcal{N}^0 \cup C, \mathcal{E}^1)$. Specifically, for each query concept $q \in C$, the task is to find suitable positions in $\mathcal{T}^0$, identified by candidate parent-child pairs $\langle p, c \rangle$, into which $q$ can be inserted. Following the assumptions of prior works [39, 70], the problem is decomposed into $|C|$ independent insertion tasks, where $|C|$ is the number of query concepts.

## 3.2 Taxonomic Semantic Compression

The goal of taxonomic semantic compression is to convert the long concept description input to a single token that LLM can understand and use. To achieve this, we leverage the LLM to generate concept representations directly within its own semantic space, ensuring seamless understanding when these representations are reintroduced to the LLM (§3.2.1). Then we input these tokens to the LLM for a task-specific compression objective, preserving taxonomy-related semantics in the compressed tokens (§3.2.2).

*3.2.1 **LLM-based Concept Representation Generation**.* To compress concept descriptions and generate representations aligned with the semantic space of an LLM, we adopt a direct approach using the LLM itself, ensuring natural compatibility. Unlike existing methods that align external representations with the LLM's space [16, 34], we simplify the process by generating concept representations directly through the LLM. Specifically, we follow the approach of PromptEOL [38], whose "one-word limitation" aligns with our compression objective, to generate representations. Given a concept description $d_n$, we use the prompt function $\mathcal{F}_{con}(d_n)$ to query the LLM:

> *Please summarize the meaning of concept description: <$d_n$> in one word:*

After autoregressive decoding, the hidden vector following "in one word:" is extracted as the concept representation, denoted as $h_n$.

*3.2.2 **Taxonomy-Specific Compression Task**.* To embed task-relevant semantics into the generated representations, we design a taxonomy-related compression task where the LLM is used to complete taxonomy by treating concept representations as input tokens. Following [31], given a query node $q$ and a candidate position $\langle p, c \rangle$, we extend the candidate position to $\langle p, c, s \rangle$ by randomly selecting a sibling $s$ (a child of $p$). The representations $h_q, h_p, h_c, h_s$ are generated as outlined in Section 3.2.1. We then apply the prompt function $\mathcal{F}_{pos}(h_p, h_c, h_s)$ to provide $h_p, h_c, h_s$ as input tokens:

> *I'm finding a target concept, whose parent concept is: <$h_p$>, child concept is: <$h_c$>, and sibling concept is <$h_s$>. Please predict the meaning of the target concept in one word:*

The hidden vector following "in one word:" becomes the representation for the candidate position, denoted as $h_{pos}$. We train the model using BCELoss:

$$\mathcal{L}_{comp} = -\log \sigma \left( h_q \cdot h_{pos}^+ \right) - \sum_{i=1}^{NS} \log \sigma \left( 1 - h_q \cdot h_{pos}^{-,i} \right), \quad (1)$$

where $NS$ is the number of negative samples, and $h_{pos}^{-,i}$ denotes the $i$-th negative sample. $\sigma$ represents the sigmoid function.

As depicted in Figure 2, this task jointly trains the LLM for *Concept Representation Generation* and *Taxonomy Completion*. However, this approach requires significant GPU memory and computation time, limiting the integration of structural information. To address this, we adopt a two-stage process: after the first stage, where the compression task converges, we **freeze the concept representations and store them in a look-up table**. In the second stage, we retrieve these precomputed representations to focus on training the LLM for structure modeling.

## 3.3 Contrastive Structure Modeling

After compression, the model processes longer path sequences to capture structural information. We apply contrastive learning to differentiate these sequences. Given a candidate position $\langle p, c, s \rangle$, we construct three path sequences: (i) $\mathcal{S}(p)$ traces the longest path from the root to parent $p$, capturing the "is-a" relationship [24, 57]; (ii) $\mathcal{S}(c)$ extends from child $c$ to the leaf, also the longest; (iii) $\mathcal{S}(s)$ samples all siblings of $s$, derived from the children of $p$ [12] with a fixed alphabetical order. Each node in the path is represented by its compressed token, preserving the structural clarity. Using the prompt function $\mathcal{F}_{strc}(\mathcal{S}(p), \mathcal{S}(c), \mathcal{S}(s))$, we query the LLM to predict the target concept's meaning, extracting the hidden vector $h_{strc}$ as the structure representation:

> *I'm finding a target concept, whose parent concepts from general to specific are: <$\mathcal{S}(p)$>, child concepts from general to specific are: <$\mathcal{S}(c)$>, and sibling concepts are: <$\mathcal{S}(s)$>. Please predict the meaning of target concept in one word:*

By using the freezed concept representations, we reduce GPU memory usage, allowing more negative samples for contrastive

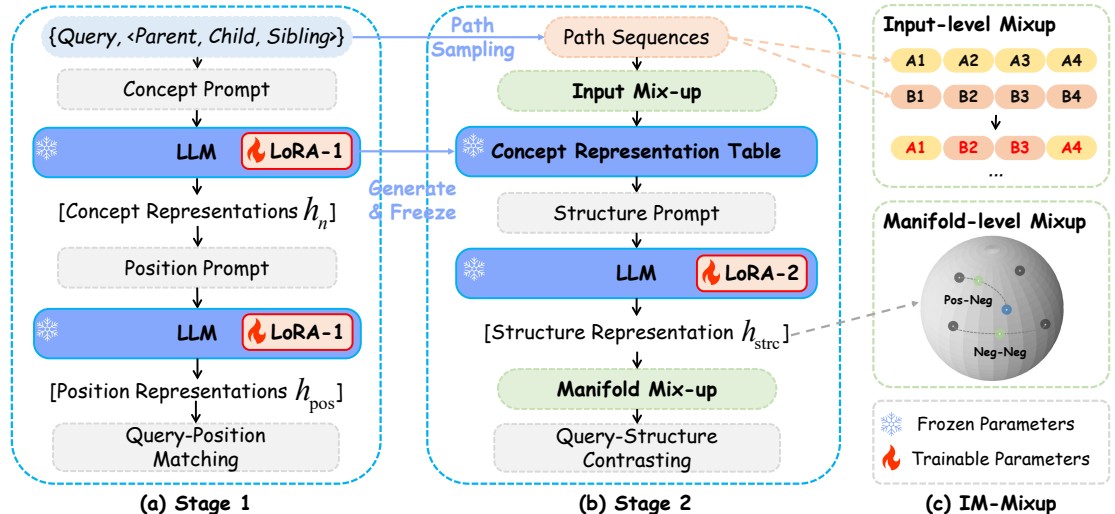

**Figure 2: Illustration of our framework.**

learning. Our contrastive objective is defined as:

$$\mathcal{L}_{\text{strc}} = (d(h_q, h_{\text{strc}}^+))^2 + \sum_{i=1}^{RS} (max(0, m - d(h_q, h_{\text{strc}}^{-,i})))^2, \quad (2)$$

where $m \in (0, 1)$ is a margin hyperparameter, $RS$ is the number of negative samples, and $d(h_q, h_{\text{strc}})$ is the cosine distance between query $h_q$ and structure $h_{\text{strc}}$. All representations are $L_2$-normalized for stable learning [58]. The choice of contrastive loss for this stage and its comparison to BCELoss in the previous stage are discussed in Section 4.2.3. For prompt choice details, see Appendix B.

### 3.4 Mixup Enhanced Structure Discrimination

To enhance the model's ability to uncover structural information within taxonomies, we improve our contrastive learning framework with a mixup data augmentation strategy. This approach generates diverse and challenging sequences, promoting robust representation learning. We apply mixup at two levels: input and manifold. At the input level, cut-based mixup replaces partial sequences to better capture local structures. At the manifold level, linear mixup synthesizes instances with varying difficulty, enabling finer discrimination in the feature space. This dual-level augmentation enhances the model's ability to learn sequences from multiple perspectives.

*3.4.1 **Principles for Effective Mixup**.* Given a query and its corresponding positive and negatives, we follow these principles to ensure the generated mixup samples are diverse and challenging:

- **Only hard negatives are selected for mixup.** Mixup involves linear combinations, and only hard samples within the margin contribute to the loss. By mixing samples within this margin, we ensure that the new synthesized samples also contribute to effective training.
- **Assign larger mixing weights to similar negatives.** Based on the findings in [71], assigning higher mixing weights to more similar negatives generates more discriminative negative pairs.

- **Mix positives with hard negatives for more challenging negatives.** As suggested in [15], mixing hard negatives alone does not always yield harder negatives since the created hard negatives lie inside the convex hull of the hard negatives. To address this, we mix positives with negatives for more challenging instances. By setting the positive's mixing weight below 0.5, we ensure the negative sample remains dominant. We term the negative-negative mix as *neg-neg*, producing *hard* samples, and the positive-negative mix as *pos-neg*, which generates *harder* samples.

*3.4.2 **I-Mix: Input-level Cut-based Mixup**.* To help the model capture sequence patterns and subtle local structures, we apply cut-based mixup by randomly replacing nodes in one path with nodes from another, generating a mixed path. We define the path sequence $\mathbf{P}$ consisting of $H$ nodes as: $\mathbf{P} = \mathcal{S}(p) \circ \mathcal{S}(c) \circ \mathcal{S}(s)$, where $\circ$ denotes the tensor concatenation of the node embeddings $h \in \mathbb{R}^{1 \times D}$. Given two path sequences, $\mathbf{P}_i$ and $\mathbf{P}_j$, we define the combining operation as:

$$\hat{\mathbf{P}} = \mathbf{M} \odot \mathbf{P}_i + (1 - \mathbf{M}) \odot \mathbf{P}_j, \quad (3)$$

where $\mathbf{M} \in \{0, 1\}^{1 \times (H \cdot D)}$ is a binary mask indicating cut-and-paste areas, and $\odot$ is element-wise multiplication. The mask applies to whole nodes, meaning each node's embedding is either fully included or excluded to prevent splitting and noise. For *neg-neg* mix, the masking ratio $\alpha$ is determined by the overlap ratio between the sequence $\mathbf{P}_i$ and the ground-truth path $\mathbf{P}^+$, quantified as:

$$\alpha_i = \frac{\exp(\mathcal{H}(\mathbf{P}_i, \mathbf{P}^+))}{\exp(\mathcal{H}(\mathbf{P}_i, \mathbf{P}^+)) + \exp(\mathcal{H}(\mathbf{P}_j, \mathbf{P}^+))}, \quad (4)$$

where $\mathcal{H}$ is the similarity function defined as the overlap ratio. For *pos-neg* mix, the masking ratio $\alpha$ is sampled from the uniform distribution $(0, 1)$ [15, 65]. We enforce that a sequence is only considered correct if all nodes in the path are accurate, rather than just directly connected nodes, pushing the model to learn fine-grained local structural differences.

*3.4.3* **M-Mix: Manifold-level Linear Mixup**. To enhance global discrimination of path sequences, we perform manifold-level linear mixup. Studies [4, 48, 49] have demonstrated that linear interpolation in the embedding space better addresses decision boundary issues and provides greater sample diversity than input-level mixup, introducing more structural perturbation in embedding space. For a pair of path sequence representations $h_i$ and $h_j$, we define their convex combination as:

$$\hat{h} = \lambda h_i + (1 - \lambda) h_j, \tag{5}$$

where $\lambda \in (0, 1)$ is the mixing weight. To simplify notation, subscripts on $h_{\text{strc}}$ are omitted. Since mixing forms a linear combination of embeddings, the synthesized samples lie along the line segments connecting the original pairs, ignoring the effects of $L_2$-normalization for this analysis.

For *neg-neg* mix, the mixing weight $\lambda$ is determined by:

$$\lambda = \frac{\exp(\mathcal{H}(h_i, h^+))}{\exp(\mathcal{H}(h_i, h^+)) + \exp(\mathcal{H}(h_j, h^+))}, \tag{6}$$

where $\mathcal{H}$ is the cosine similarity function. Following [48, 71], we extend the mixup process to the entire batch to increase sample diversity. For each sample, the mixing weight is calculated as:

$$\lambda_i = \frac{\exp(\mathcal{H}(h_i, h^+))}{\sum_i \exp(\mathcal{H}(h_i, h^+))}, \quad s.t. \quad i \in [0, RS], \tag{7}$$

where $RS$ indicates the random negative size in sampling. The generated new sample $\hat{h}$ becomes: $\hat{h} = \sum_i^{RS} \lambda_i h_i$.

For *pos-neg* mix, our primary focus is on the diversity of synthesized samples, as those mixed with positive samples already provide sufficient information. Given that contrastive learning aims to separate positive and negative samples in the embedding space, we prioritize maximizing directional diversity within this space. Our objective is to refine decision boundaries in multiple directions using a minimal number of samples. To this end, for each positive sample $h^+$, we select representative negative samples $h_k$ that span distinct directions relative to $h^+$ with a random mixing weight $\lambda \in (0.5, 1)$. We employ a greedy strategy to iteratively choose negative vectors that maximize angular distance from the previously selected directions. The angular distance between $h_i$ and $h_j$ is defined as follows:

$$\theta_{i,j} = \arccos\left(\left(\frac{h_i - h^+}{||h_i - h^+||}\right) \cdot \left(\frac{h_j - h^+}{||h_j - h^+||}\right)\right). \tag{8}$$

For a visual understanding of the M-Mix strategy, please see Figure 6 in the Appendix A.

Finally, we utilize the mixtures as additional new entrees of contrastive loss. The number of mixed samples $MS$ is determined through experiments. To ensure stable training and optimal performance [9, 43], we set a ratio $r$ of hard samples to total samples. The effects of these hyperparameters are discussed in Section 4.2.3. Our IM-Mix performs mixup operations in both the input space and the representation space. Since the input is also a tensor, the mixup operation is computationally efficient and thus creates query-specific synthetic points on the fly. The synthesized samples are informative and able to show improved results at a smaller number of epochs [15], as shown in Figure 7 in the Appendix B.

**Table 1: The dataset statistics. $|\mathcal{N}|$ and $|\mathcal{E}|$ represent the total number of nodes and edges, respectively. The terms #depth and #avg.tokens refer to the taxonomy's depth and the description's average token length.**

| Dataset | $|\mathcal{N}|/|\mathcal{N}_{\text{train}}|$ | $|\mathcal{E}|$ | #depth | #avg.tokens | #candidates |
|---|---|---|---|---|---|
| SemEval-Food | 1486/1190 | 1,533 | 8 | 34.6 | 7313 |
| MeSH | 9710/8072 | 10,498 | 10 | 62.6 | 42970 |
| WordNet-Verb | 13936/11936 | 13,407 | 12 | 26.4 | 51159 |

## 4 Experiments

### 4.1 Experimental Setup

*4.1.1* **Datasets.** Following [52, 57], we evaluate our method on three taxonomy completion datasets: **SemEval-Food**, which features a food domain taxonomy derived from SemEval-2015 Task 17 [3]. **Medical Subject Headings (MeSH)**, that consists of a widely used clinical domain taxonomy, serving as a subgraph of the Medical Subject Headings [19], which is a hierarchy for biomedical indexing. **WordNet-Verb**, which contains a verb taxonomy derived from SemEval-2016 Task 14 [14], representing a hierarchy of verbs from WordNet 3.0. For each taxonomy, we partition nodes $\mathcal{N}$ into non-overlapping train nodes $\mathcal{N}_{\text{train}}$, validation nodes $\mathcal{N}_{\text{validation}}$ and test nodes $\mathcal{N}_{\text{test}}$ [52, 57]. Specifically, for WordNet-Verb, we randomly sample 1,000 nodes for validation and test sets. For SemEval-Food and MeSH, we allocate 10% of the nodes as validation and another 10% as test nodes. The remaining nodes constitute the training set $\mathcal{N}_{\text{train}}$. Table 1 provides statistical information on three datasets.

*4.1.2* **Evaluation Metrics.** Following previous work [52, 57, 70], we adopt the all-rank evaluation protocol. We utilize several metrics for performance evaluation, including Macro Mean Rank (**MR**), the scaled Mean Reciprocal Rank (**MRR**) [39], **Recall@**$k$, and **Hit@**$k$.

*4.1.3* **Baseline Methods and Implementation Details.** Our method falls into **representation-based** approach for taxonomy completion. We begin by comparing it to state-of-the-art methods in this category, including TMN [70], TaxoEnrich [12], QEN [52], TaxoComplete [1], and CoSTC [31]. Since no existing baseline utilizes LLMs, we modify CoSTC, the most competitive representation-based method, by replacing its backbone with the same LLM we utilize for comparison, naming it **CoSTC-LLaMA**. Following prior work [57, 70], we adapt taxonomy expansion baselines like Taxo-Expan [39] and Arborist [26] to the taxonomy completion task by concatenating the parent and child node representations to form the candidate position's representation. However, the generation-based methods, such as TaxoLlama [29], are unsuitable for taxonomy completion due to their focus on unidirectional parent-query relationships, whereas bidirectional relationships are required. To further evaluate our method's performance, we also compare it to leading **interaction-based** techniques like TEMP [24] and Taco-Prompt [57]. Details on baseline and implementation are provided in the Appendix A.1 and A.2, respectively.

### 4.2 Experimental Results

*4.2.1* **Comparison With Baselines.** Table 2 presents a comparison of COMI's performance against various baseline methods across different scale datasets, SemEval-Food, MeSH, and WordNet-Verb.

**Table 2: Overall results on three datasets. ↓ means the lower value is better. †: interaction-based baselines. The best and second results are in bold and underlined, respectively. "Leaf" and "Non-leaf" indicate whether the query's correct insertion is as a leaf node or an intermediate node. For comparison, we replace the backbone from LLM to the PLM, i.e. BERT [7].**

| Datesets | Methods | Total | | | | | | | | Leaf | | | Non-leaf | | |
|---|---|---|---|---|---|---|---|---|---|---|---|---|---|---|---|
| | | MR↓ | MRR | R@1 | R@5 | R@10 | H@1 | H@5 | H@10 | MRR | H@5 | R@10 | MRR | H@5 | R@10 |
| SemEval-Food | TaxoExpan | 371.291 | 0.286 | 5.7 | 13.3 | 18.0 | 11.5 | 26.4 | 34.5 | 0.477 | 30.1 | 35.6 | 0.130 | 8.0 | 3.6 |
| | Arborist | 256.491 | 0.290 | 13.0 | 18.0 | 21.0 | 26.4 | 34.5 | 38.5 | 0.466 | 39.0 | 38.5 | 0.146 | 12.0 | 6.7 |
| | TMN | 173.516 | 0.332 | 10.7 | 18.7 | 22.0 | 21.6 | 36.5 | 39.9 | 0.538 | 41.5 | 41.5 | 0.164 | 12.0 | 6.1 |
| | TaxoEnrich | 230.424 | 0.408 | 11.7 | 26.7 | 31.7 | 23.6 | 49.3 | 58.1 | 0.723 | 58.5 | 66.7 | 0.149 | 4.0 | 3.0 |
| | QEN | 336.554 | 0.439 | 21.9 | 30.9 | 35.0 | 45.9 | 58.8 | 64.9 | 0.732 | 64.2 | 68.9 | 0.209 | 32.0 | 9.1 |
| | TaxoComplete | 296.072 | 0.489 | 14.7 | 30.0 | 38.0 | 29.7 | 55.4 | 65.5 | 0.702 | 60.2 | 65.2 | 0.315 | 32.0 | 15.8 |
| | CoSTC | 61.471 | 0.658 | 18.7 | 43.0 | 54.3 | 39.0 | 73.4 | 80.4 | 0.825 | 74.5 | 78.0 | 0.529 | 68.0 | 36.0 |
| | CoSTC-LLaMA | 41.457 | 0.674 | 22.2 | 46.0 | 55.3 | 46.6 | 84.5 | 87.8 | 0.934 | 89.4 | 90.4 | 0.475 | 60.0 | 28.4 |
| | TEMP† | 51.374 | 0.579 | 20.3 | 41.2 | 47.9 | 42.6 | 76.4 | 81.1 | 0.881 | 81.3 | 83.0 | 0.348 | 52.0 | 21.0 |
| | TacoPrompt† | 47.423 | 0.708 | 30.9 | 51.1 | 60.1 | 64.9 | 85.8 | 86.5 | 0.899 | 87.8 | 87.4 | 0.561 | 76.0 | 39.2 |
| | COMI-LLaMA | 25.321 | 0.724 | 28.9 | 52.1 | 61.7 | 60.8 | 87.8 | 93.2 | 0.945 | 89.4 | 92.5 | 0.555 | 80.0 | 38.1 |
| | COMI-BERT | 161.094 | 0.581 | 21.9 | 38.9 | 46.9 | 45.9 | 68.2 | 75.0 | 0.818 | 74.0 | 79.3 | 0.399 | 40.0 | 22.2 |
| MeSH | TaxoExpan | 1029.344 | 0.233 | 2.7 | 6.2 | 12.2 | 6.0 | 12.7 | 23.9 | 0.381 | 16.3 | 24.3 | 0.137 | 5.0 | 4.3 |
| | Arborist | 843.199 | 0.337 | 5.0 | 13.6 | 21.8 | 11.0 | 25.8 | 37.4 | 0.437 | 26.7 | 30.6 | 0.271 | 23.8 | 16.0 |
| | TMN | 567.831 | 0.372 | 7.2 | 17.3 | 24.6 | 15.9 | 33.6 | 43.8 | 0.525 | 38.4 | 40.7 | 0.271 | 23.4 | 14.1 |
| | TaxoEnrich | 393.062 | 0.424 | 7.4 | 22.4 | 31.0 | 16.2 | 42.6 | 52.5 | 0.619 | 51.3 | 54.1 | 0.296 | 24.1 | 15.9 |
| | QEN | 451.253 | 0.438 | 7.5 | 21.3 | 30.8 | 17.1 | 43.1 | 55.9 | 0.611 | 51.1 | 51.8 | 0.332 | 26.1 | 17.9 |
| | TaxoComplete | 357.494 | 0.540 | 10.8 | 29.3 | 41.1 | 24.5 | 54.1 | 63.9 | 0.605 | 53.8 | 52.5 | 0.500 | 54.8 | 34.1 |
| | CoSTC | 109.081 | 0.600 | 11.0 | 34.6 | 47.5 | 24.9 | 61.5 | 72.6 | 0.741 | 63.5 | 66.7 | 0.512 | 57.4 | 35.7 |
| | CoSTC-LLaMA | 47.617 | 0.672 | 12.9 | 39.9 | 54.3 | 29.4 | 72.3 | 82.7 | 0.822 | 73.7 | 74.2 | 0.579 | 69.3 | 41.9 |
| | TEMP† | 80.291 | 0.612 | 13.8 | 35.3 | 48.0 | 31.4 | 66.5 | 77.5 | 0.839 | 75.4 | 77.6 | 0.471 | 47.5 | 29.8 |
| | TacoPrompt† | 49.140 | 0.674 | 17.9 | 42.4 | 55.9 | 40.7 | 74.6 | 84.6 | 0.868 | 79.0 | 81.5 | 0.554 | 65.1 | 40.2 |
| | COMI-LLaMA | 29.477 | 0.727 | 19.9 | 47.6 | 61.5 | 45.3 | 79.4 | 88.5 | 0.855 | 80.6 | 88.7 | 0.648 | 76.6 | 50.3 |
| | COMI-BERT | 140.903 | 0.600 | 15.1 | 35.8 | 46.9 | 34.4 | 64.8 | 72.9 | 0.741 | 69.2 | 67.2 | 0.513 | 55.6 | 34.4 |
| WordNet-Verb | TaxoExpan | 1752.271 | 0.215 | 4.1 | 11.4 | 15.1 | 6.1 | 17.1 | 22.5 | 0.354 | 20.5 | 26.7 | 0.057 | 3.1 | 1.7 |
| | Arborist | 1455.251 | 0.246 | 3.8 | 11.0 | 15.5 | 5.7 | 15.5 | 21.6 | 0.331 | 16.2 | 21.8 | 0.148 | 12.8 | 8.4 |
| | TMN | 1513.634 | 0.290 | 5.4 | 14.7 | 20.7 | 8.1 | 21.2 | 29.1 | 0.425 | 23.8 | 32.8 | 0.136 | 10.7 | 6.8 |
| | TaxoEnrich | 5462.075 | 0.179 | 3.9 | 9.0 | 12.3 | 5.8 | 13.6 | 18.4 | 0.313 | 16.8 | 22.6 | 0.025 | 0.5 | 0.4 |
| | QEN | 1730.755 | 0.404 | 9.1 | 23.3 | 31.0 | 13.9 | 34.0 | 43.9 | 0.568 | 38.6 | 48.4 | 0.224 | 15.3 | 11.8 |
| | TaxoComplete | 2661.488 | 0.407 | 9.0 | 22.2 | 30.9 | 13.6 | 31.7 | 40.8 | 0.487 | 32.7 | 41.3 | 0.315 | 27.6 | 19.1 |
| | CoSTC | 241.849 | 0.505 | 9.5 | 27.8 | 39.1 | 14.6 | 39.2 | 53.1 | 0.651 | 41.0 | 54.7 | 0.344 | 31.6 | 21.8 |
| | CoSTC-LLaMA | 176.405 | 0.545 | 15.6 | 34.3 | 43.1 | 23.9 | 51.0 | 60.6 | 0.727 | 55.0 | 63.6 | 0.346 | 34.7 | 20.5 |
| | TEMP† | 960.536 | 0.450 | 13.3 | 30.6 | 37.5 | 20.3 | 45.9 | 55.0 | 0.692 | 53.4 | 62.8 | 0.182 | 15.3 | 9.5 |
| | TacoPrompt† | 436.799 | 0.557 | 18.3 | 36.9 | 46.5 | 28.0 | 52.3 | 62.5 | 0.762 | 56.5 | 65.8 | 0.370 | 35.2 | 25.3 |
| | COMI-LLaMA | 109.454 | 0.615 | 19.3 | 39.9 | 50.6 | 29.6 | 55.5 | 66.5 | 0.760 | 58.5 | 68.0 | 0.455 | 43.4 | 31.3 |
| | COMI-BERT | 478.972 | 0.500 | 15.3 | 33.3 | 41.1 | 23.4 | 47.0 | 55.1 | 0.652 | 51.1 | 58.8 | 0.333 | 30.1 | 21.6 |

**Table 3: Inference time (in minutes) comparison of different settings. All methods are tested using the maximum inference batch size on a single A800-80G GPU.**

| Settings | SemEval-Food | MeSH | WordNet-Verb |
|---|---|---|---|
| TacoPrompt | 23.7 | 940.5 | 1193.8 |
| Ours (w/o comp) | 3.2 | 36.0 | 15.7 |
| Ours (w/ comp) | 0.4 (59.3×) | 3.7 (254.2×) | 4.2 (284.2×) |

Table 3 compares the inference efficiency of COMI with the current SOTA method, TacoPrompt. We discuss the question below.
**Q1. How effective and efficient is COMI for taxonomy completion?** In terms of effectiveness, COMI achieves significant improvements within the representation-based taxonomy completion task. We replace the backbone model of the previous SOTA method, CoSTC, with the same LLM used in our approach. With both methods using LLaMA, COMI achieves **absolute improvements in**

**Hit@1 by 14.2%, 15.9%, and 5.1%** on the SemEval-Food, MeSH, and WordNet-Verb datasets, respectively, demonstrating its ability to effectively leverage the semantic and structure modeling capabilities of the LLM. Compared to interaction-based methods such as TEMP and TacoPrompt, our approach performs comparably on the SemEval-Food dataset and significantly outperforms them on MeSH and WordNet-Verb. These results highlight the superior performance of our approach. From an efficiency perspective, interaction-based methods are constrained by high inference costs, which limit their use of LLMs. In contrast, COMI achieves up to **284× faster inference** than TacoPrompt, providing the optimal balance between performance and efficiency.

*4.2.2 Ablation Studies.* We conduct ablation studies on key components of the semantic compression and structure modeling stages to explore the following questions.
**Q2. What is the function of the first compression stage?** Table 4 shows ablation results for the first semantic compression

**Table 4: Ablation studies of the semantic compression stage. We compare all settings w/o IM-Mix, as it is specifically designed for the compressed inputs. #Neg and #TT represent the negative sampling size and training time per epoch (in minutes) on a single A800-80G GPU, respectively. Due to GPU memory limitations without compression, we compare different settings where #Neg is set to the maximum possible for the non-compressed setting and matched to our approach in the compressed setting. For the "w/o comp task", we generate concept representations in a zero-shot manner. For a detailed discussion on the effects of different compression tasks, please refer to Appendix B. "w/o comp" means we utilize descriptions instead of compressed tokens as LLM's input and "w/o struct" means we only use position rather than path sequences.**

| Settings | #Neg | #TT | MRR | H@1 | H@5 | R@5 | R@10 |
|---|---|---|---|---|---|---|---|
| **SemEval-Food** | | | | | | | |
| Ours | 40 | **8.5** | **0.716** | **56.8** | **85.8** | **50.2** | **60.1** |
| w/o comp task | 40 | **8.5** | 0.531 | 37.8 | 69.6 | 36.3 | 41.2 |
| w/o comp & struct | 40 | **32.3** | 0.701 | 52.0 | 85.8 | 47.3 | 58.8 |
| Ours | 20 | **3.5** | 0.708 | 54.5 | 82.4 | 48.6 | 59.2 |
| w/o comp | 20 | **32.4** | 0.704 | 53.4 | 85.1 | 47.3 | 57.8 |
| w/o comp & struct | 20 | **14.7** | 0.689 | 56.1 | 85.1 | 46.9 | 57.2 |
| **MeSH** | | | | | | | |
| Ours | 40 | **58.3** | **0.702** | **42.6** | **77.7** | **45.4** | **58.8** |
| w/o comp task | 40 | **58.3** | 0.521 | 17.9 | 54.5 | 27.5 | 38.8 |
| w/o comp & struct | 40 | **696.5** | 0.675 | 32.7 | 73.9 | 41.3 | 55.1 |
| Ours | 10 | **16.3** | 0.680 | 41.4 | 76.7 | 43.6 | 56.3 |
| w/o comp | 10 | **237.5** | 0.675 | 23.0 | 73.8 | 41.2 | 55.1 |
| w/o comp & struct | 10 | **137.7** | 0.665 | 33.7 | 72.3 | 39.9 | 54.3 |
| **WordNet-Verb** | | | | | | | |
| Ours | 40 | **79.0** | **0.606** | **27.1** | **51.1** | **37.3** | **48.6** |
| w/o comp task | 40 | **79.0** | 0.420 | 10.8 | 36.1 | 23.9 | 32.3 |
| w/o comp & struct | 40 | **226.0** | 0.580 | 17.3 | 50.3 | 34.9 | 46.7 |
| Ours | 15 | **28.2** | 0.578 | 25.3 | 49.6 | 35.1 | 45.9 |
| w/o comp | 15 | **149.6** | 0.558 | 20.5 | 50.5 | 34.2 | 44.1 |
| w/o comp & struct | 15 | **81.3** | 0.545 | 24.5 | 50.2 | 33.7 | 43.6 |

stage. We observe the following: (1) the taxonomy-related compression task outperforms zero-shot semantic compression, ensuring that the compressed representations capture task-relevant semantic knowledge; (2) omitting compression significantly reduces training efficiency, with the slowest experiment taking approximately 15 days training on an A800 GPU; (3) without compression, the model struggles to integrate both semantic and structural information, as concatenating concept descriptions with "\n" in the "w/o comp" setting fails to preserve the taxonomic hierarchy, resulting in lower Hit@1 performance compared to the "w/o comp & struct" setting; and (4) using compression effectively integrates semantic and structural information, achieving the best efficiency and performance across different negative sampling rates.

**Q3. How effective are the design choices in stage two of structure modeling?** Table 5 presents the ablation results for the second stage of structure modeling, evaluating the contributions of IM-Mix data augmentation, path sequence usage, and contrastive learning. For **IM-Mix data augmentation**, we sequentially removed

**Table 5: Ablation studies of the structure modeling stage. A detailed analysis of the effects of different path sequence components is provided in Appendix B.**

| Datasets | Settings | MRR | H@1 | H@5 | R@5 | R@10 |
|---|---|---|---|---|---|---|
| SemEval-Food | Ours | **0.724** | **60.8** | **87.8** | **52.1** | **61.7** |
| | w/o I-Mix | 0.718 | 58.7 | 85.8 | 50.8 | 60.1 |
| | w/o IM-Mix | 0.716 | 56.8 | 85.8 | 50.2 | 60.1 |
| | w/o IM-Mix & struct | 0.693 | 58.8 | 83.1 | 47.6 | 58.5 |
| | w/o stage two | 0.699 | 53.4 | 84.5 | 45.0 | 56.3 |
| MeSH | Ours | **0.727** | **45.3** | **79.4** | **47.6** | **61.5** |
| | w/o I-Mix | 0.710 | 43.6 | 78.9 | 46.3 | 58.9 |
| | w/o IM-Mix | 0.702 | 42.6 | 77.7 | 45.4 | 58.8 |
| | w/o IM-Mix & struct | 0.684 | 40.8 | 74.0 | 43.0 | 56.6 |
| | w/o stage two | 0.688 | 20.0 | 72.8 | 38.5 | 55.4 |
| WordNet-Verb | Ours | **0.615** | **29.6** | **55.5** | **39.9** | **50.6** |
| | w/o I-Mix | 0.609 | 28.1 | 53.9 | 38.2 | 50.2 |
| | w/o IM-Mix | 0.606 | 27.1 | 51.1 | 37.3 | 48.6 |
| | w/o IM-Mix & struct | 0.567 | 25.5 | 48.9 | 34.2 | 44.7 |
| | w/o stage two | 0.570 | 22.2 | 49.6 | 33.6 | 44.3 |

**Table 6: Comparison of the semantic knowledge in representations. We leverage TMN as the backbone model, whose original version utilized fixed fastText [2] representations. MRR metric is used for comparison.**

| Representations | SemEval-Food | MeSH | WordNet-Verb |
|---|---|---|---|
| fastText | 0.332 | 0.372 | 0.290 |
| LLaMA-Zero-Shot | 0.512 | 0.514 | 0.416 |
| Ours-BERT | 0.595 | 0.604 | 0.475 |
| Ours-LLaMA | **0.650** | **0.666** | **0.555** |

input Mix (I-Mix) and manifold mix (M-Mix), denoted as w/o I-Mix and w/o IM-Mix, respectively. The latter indicates the additional removal of M-Mix after I-Mix. To assess the importance of **structural information**, we further ablated the path sequence (w/o IM-Mix & struct), which also necessitated the removal of mixup, as it was designed for path sequences. Finally, we eliminated the entire **stage two** training process, forgoing contrastive learning. This primarily impacted the @1 metric, highlighting the model's diminished capacity for fine-grained distinctions when trained solely with BCELoss. This is due to the increased negative sample size after compression in the first stage, which proved to be essential for contrastive learning. Each module's removal resulted in a performance drop, underscoring their effectiveness in the overall model.

*4.2.3* ***Further Discussions.*** Our further discussions include: (i) the motivation demonstration of using LLMs for the taxonomy completion task (**Q4**), (ii) the effects of training objectives for two stages (**Q5**), (iii) the impact of key hyperparameters: random negative sample number $RS$, mixup sample number $MS$ and hard mixup sampling ratio $r$ (**Q6, Q7**), and (iv) the mixup visualization (**Q8**).

**Q4. Is LLM a good choice for semantic knowledge and structure modelling for the taxonomy completion task?** Table 6 compares the semantic knowledge compressed by LLaMA with other representations, showing that our approach captures more taxonomy-relevant information, providing a foundation for future research. Figure 3 (b) evaluates the ability of various models to leverage structural information in the structure modeling stage, with

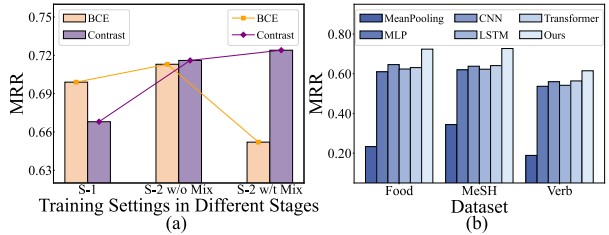

Figure 3: (a) Choice of training objectives for two stages on SemEval-Food. S-1 represents the semantic compression stage and S-2 refers to the structure modeling stage. (b) Comparison of two-stage structure modeling methods.

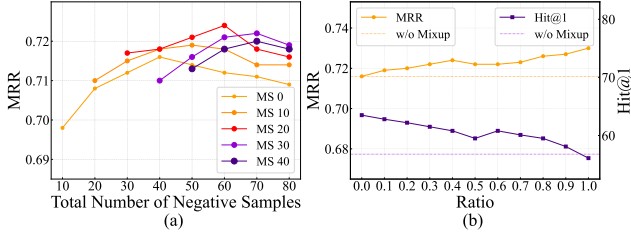

Figure 4: (a) The performance of our method on the SemEval-Food dataset with varying total negative sample sizes, defined as the sum of random negatives size $NS$ and mixup samples size $MS$. (b) Sensitivity analysis of the mixup sampling ratio $r$ hyperparameter on the SemEval-Food dataset.

LLaMA outperforming others in comprehension. Finally, as shown in Table 2, when replacing LLaMA with BERT, LLaMA demonstrates superior integration of semantic and structural knowledge, justifying its use in the taxonomy completion task.

**Q5. What are effects of training objectives for two stages?** As illustrated in Figure 3 (a), using BCELoss in the first stage outperforms contrastive loss, primarily due to GPU memory limitations that hinder the use of an adequate negative sampling rate required for contrastive learning. In the second stage, freezing the concept representations from the compression stage alleviates these memory constraints, allowing for the effective application of contrastive loss [31]. Furthermore, our proposed mixup method, tailored for contrastive learning, results in a performance decline when used with BCELoss in the second stage. Therefore, we opt for BCELoss in the first stage and contrastive loss in the second stage.

**Q6. What is the impact of different combinations of random negative sample number $RS$ and mixup sample number $MS$?** We investigate different combinations of random negative samples $RS$ and mixup samples $MS$ as shown in Figure 4 (a), leading to three key observations. First, the number of random samples $RS$ should not exceed 50, as higher values result in performance degradation across different $MS$ values, due to overfitting to simple features, which hinders fine-grained path sequence distinction. Second, the ratio between $RS$ and $MS$ requires careful balancing. For instance, $RS = 10, MS = 30$ performs worse than not using mixup, i.e., $RS = 40, MS = 0$. Lastly, with the same total number of samples, using mixup improves performance when the $RS$-$MS$ ratio is optimal.

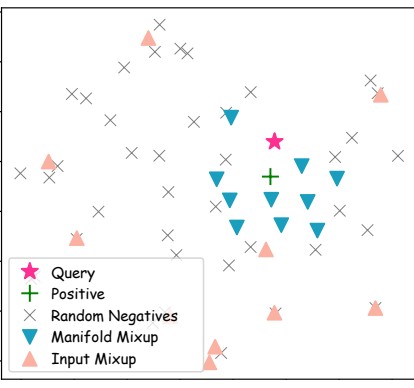

Figure 5: t-SNE [47] representations of positive, random negatives, and our mixup negatives for the concept "wild rice". Note that IM-Mix generates synthetic diverse and challenging negatives for each query.

For instance, $RS = 40, MS = 20$ outperforms $RS = 60, MS = 0$, highlighting the effectiveness of mixup over merely increasing random negative samples.

**Q7. How sensitive is our framework to the mixup sampling ratio $r$?** The mixup sampling ratio $r$ controls the balance between moderately hard (*neg-neg*) and harder (*pos-neg*) samples. As shown in Figure 4 (b), the model remains robust when $r$ is between 0.3 and 0.7. A lower $r$ increases harder samples, resulting in a lower MRR but higher Hit@1, while a higher $r$ has the opposite effect. Thus, a mid-range $r$ provides a more balanced performance.

**Q8. What kind of samples does IM-Mix synthesize to enhance contrastive learning?** Figure 5 presents a t-SNE visualization of the learned representation space after applying IM-Mix to a mini-batch. The query concept (red star) is surrounded by random negatives (gray marks), where many are too distant to significantly impact the contrastive loss. Negatives generated by I-Mix (pink triangles), which alters the local structure of input paths, exhibit a slight shift in embedding space. M-Mix-generated negatives (blue triangles), synthesized using hard negatives based on their similarity to the positive, are more challenging and dispersed in various directions. This demonstrates the effectiveness of our mixup strategy in producing more diverse and difficult samples.

## 5 Conclusion

In this paper, we present COMI, an efficient framework for taxonomy completion that leverages the strengths of the LLMs. COMI integrates semantic compression and contrastive learning with mixup data augmentation to address both semantic and structural challenges in taxonomy completion. The use of compressed tokens allows for efficient inference while maintaining semantic richness and structural clarity. The mixup augmentation enhances structural complexity, fostering more precise discrimination. Comprehensive experiments on real-world datasets demonstrate that COMI not only achieves SOTA performance but also significantly reduces inference time. This framework offers a promising and efficient direction for TC using LLMs and can be adapted to other knowledge-structuring tasks where both semantic and structural information are crucial.

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

## A Supplementary Details

### A.1 Baseline Introduction

The *representation-based* taxonomy completion methods include:

- **TMN** [70]: This method employs subtasks, namely attaching query to parent and child to query, as auxiliary supervision signals for concept representation learning.
- **TaxoEnrich** [12]: It utilizes structural information through taxonomy-contextualized embeddings, enhancing position representations with a query-aware sibling aggregator.

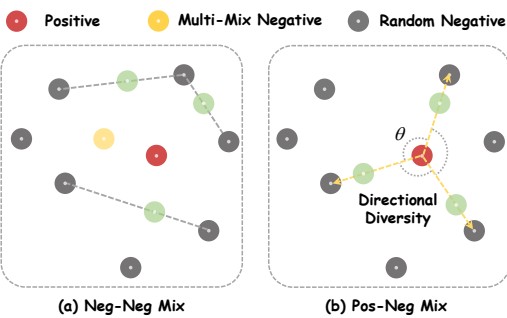

**Figure 6: A depiction of the manifold mixup strategy, where synthesized samples (green ones) are positioned along line segments connecting original data pairs. The *Pos-Neg* mix selects negatives to ensure directional diversity.**

- **QEN** [52]: This technique generates semantic concept representations using a pre-trained language model, focusing on sibling relations to mitigate pseudo-leaf noise.
- **TaxoComplete** [1]: This framework leverages semantic similarity through bi-encoders and employs direction-aware propagation for position-enhanced node representations.
- **CoSTC** [31]: This is a contrastive representation learning framework which leverages two contrastive views and a negative sampling strategy to extract taxonomic relations.

The *interaction-based* taxonomy completion techniques include:

- **TEMP** [24]: This technique calculates insertion probabilities based on the taxonomy-path, which integrates paths from the root to the parent, along with the query.
- **TacoPrompt** [57]: This method performs triplet semantic matching for taxonomy completion by combining the descriptions of parent, child, and query concepts.

Note that TEMP was originally designed for taxonomy expansion, but we use its adapted version for taxonomy completion, which attaches the child node to the taxonomy path, following [57].

### A.2 Implementation Details

We leverage LLaMA-7B [1] [46] as the backbone LLM. We train LLaMA using LoRA [10] and set its rank to 32. The model is trained using the AdamW optimizer, with a learning rate of 3e-4. Training ends if the MRR score on the validation set doesn't improve within 10 epochs. All the experiments are accelerated by an NVIDIA A800-80G GPU device. For the **first-stage** semantic compression, we sample 15 negative positions per training instance, and the batch size is set to 1. For the **second-stage** structure modeling, we load the concept representations generated by the first stage as a frozen representation as a look-up table and re-equip LLaMA with a new LoRA. The hyperparameters random negative size $RS$, mixup samples number $MS$, and the hard to total samples ratio $r$ are set to 40, 20, and 0.4 respectively, with 10 samples each for the two types of mixup. As for the contrastive loss margin $m$, calibrated on the validation set, it is set to 0.7 for SemEval-Food, 0.5 for MeSH, and 0.7 for WordNet-Verb. The batch size is set to 3. For the **backbone**

---

[1] https://huggingface.co/huggyllama/llama-7b

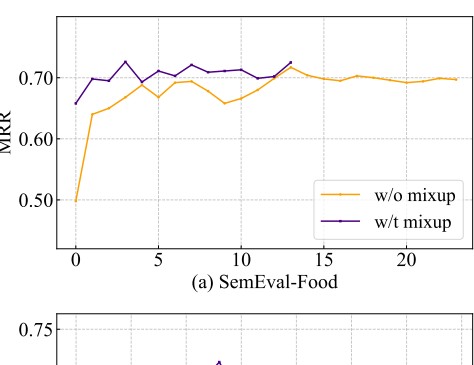

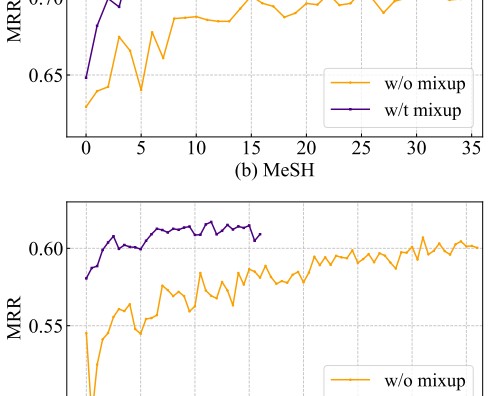

Figure 7: Effects of mixup on training convergence. The green area represents the difference in training epochs, where the epochs w/o mixup are higher compared to those w/t mixup.

Table 8: Performance of our method with different path sequence understanding prompts on SemEval-Food.

| Prompts | MRR | Hit@1 | Recall@10 |
|---|---|---|---|
| Prefix Tokens (Ours) | **0.724** | **60.8** | **61.7** |
| Position Embedding | 0.701 | 58.7 | 58.2 |
| None | 0.699 | 57.4 | 57.2 |

Table 9: Performance of our method with different path sequence components on SemEval-Food. "L" and "NL" are short for "Leaf Scenario" and "Non-Leaf Scenario", respectively.

| Settings | MRR | MRR-L | MRR-NL |
|---|---|---|---|
| Ours | **0.724** | **0.945** | **0.555** |
| w/o parent path | 0.712 | 0.920 | 0.552 |
| w/o child path | 0.711 | 0.934 | 0.541 |
| w/o sibling path | 0.709 | 0.917 | 0.551 |

Table 7: Performance of our method with different compression tasks on SemEval-Food.

| Compression Tasks | MRR | Hit@1 | Recall@10 |
|---|---|---|---|
| Ours | **0.724** | **60.8** | **61.7** |
| Hypernym-Hyponym | 0.590 | 36.5 | 47.9 |
| Unsupervised | 0.601 | 46.6 | 48.6 |
| None | 0.531 | 37.8 | 41.2 |

**discussion**, we replace LLaMA with the PLM, BERT [2] [7] and fine-tune it with a learning rate of 3e-5. For the **ablation studies**, in the "w/o comp & struct" setting, we use "\n" as a separator between the descriptions of different concepts. The negative sampling size is determined by the maximum value when each sentence within the batch is encoded individually. This minimizes memory usage and fairly highlights the significance of our compression design.

## B Supplementary Experiments

• **Effects of Different Compression Tasks.** We compared two semantic compression tasks: (1) *Hypernym-Hyponym*, trained with unidirectional hypernym and hyponym supervision, and (2) *Unsupervised*, which uses self-supervised pretraining tasks from CoSTC [31] after obtaining concept semantic and path sequence representations. Results in Table 7 demonstrate that the compression task we utilize preserves the most relevant semantic knowledge for taxonomy completion, achieving the best performance.

• **Effects of Different Prompts for LLM's Path Sequence Understanding.** We compare the explicit *Prefix Token Prompt* utilized in this paper with two alternatives: (1) *Position Embedding*, which compresses the prompt into a single embedding and treats it as a position embedding [23], which is added to the compressed token embeddings of the corresponding parent, child, and sibling path sequences, and (2) *None*, which requires the LLM to differentiate the boundaries between different path sequences without prompts. The results in Table 8 show that although the explicit prompt increases input length to some extent, it helps the LLM better understand the distinctions between path sequences.

• **Effects of Different Path Sequences.** From the results in Table 9, we observe that the parent path improves leaf insertion performance, while the child path enhances non-leaf insertion performance. Consistent with previous research [12, 31, 52], sibling information is crucial for the taxonomy completion task. Our method effectively leverages all these path sequence components, resulting in the best overall performance.

• **Effects of Mixup On Training Converge.** From the results in Figure 7, we can observe that using Mixup accelerates training convergence by the informativeness of the synthesized samples, further enhancing the efficiency of LLM training in our framework.

---

[2] https://huggingface.co/bert-base-uncased

