# OpenReview forum: "Compress and Mix: Advancing Efficient Taxonomy Completion with Large Language Models"
_ACM.org/TheWebConf/2025/Conference — WWW 2025 Poster_

### Official Review · Reviewer_z66k · 2024-11-04

**Novelty:** 5
**Technical Quality:** 5

**Review:**

**Quality:**
1. The paper presents a novel framework, COMI, for taxonomy completion that leverages large language models (LLMs) to address the challenges of integrating semantic and structural information.
2. The authors have conducted comprehensive experiments on three real-world datasets, demonstrating the effectiveness of their approach compared to existing methods.
3. The paper is well-structured, with clear definitions, methodology, experiments, and discussions.

**Clarity:**
1. The introduction provides a concise overview of the paper's contributions and findings.
2. The methodology is explained in a step-by-step manner, making it easy to follow the authors' thought process and the technical details of their approach.

**Originality:**
1. The proposed COMI framework is original in its approach to compressing semantic information and enhancing model understanding through contrastive learning and mixup data augmentation.
2. The paper introduces a new way to leverage LLMs for taxonomy completion, which is a significant departure from traditional methods that do not utilize such models.
3. The combination of semantic compression and mixup data augmentation is innovative and addresses both efficiency and accuracy in taxonomy completion.

**Significance:**
1. The work is significant as it advances the field of taxonomy completion, which is crucial for knowledge organization and information retrieval systems.
2. The paper demonstrates state-of-the-art performance while significantly improving inference speed, which is a major contribution to practical applications.
3. The framework's efficiency and flexibility could lead to broader adoption of LLMs in taxonomy completion and related knowledge structuring tasks.


**Cons:**
1. While the paper demonstrates significant improvements, it may not cover all possible applications or datasets, which could limit the generalizability of the findings.
2. The reliance on LLMs could introduce challenges related to computational resources and potential biases inherent in these models.
3. The paper could benefit from a deeper discussion on the limitations of the approach and potential future work to address these limitations.

Overall, the paper presents a high-quality, original, and significant contribution to the field of taxonomy completion. It is well-written and provides a clear explanation of the proposed framework and its evaluation.

**Questions:**

- How do COMI and the baseline compare in terms of resource dependency?
- Given the significant improvements in efficiency and performance of COMI compared to TacoPrompt, what are the disadvantages or limitations of COMI?
- What trade-offs were made to achieve these improvements?

**Reviewer Confidence:**

2: The reviewer is willing to defend the evaluation, but it is likely that the reviewer did not understand parts of the paper

**Scope:**

2: The connection to the Web is incidental, e.g., use of Web data or API

---

### Official Review · Reviewer_HWyB · 2024-11-27

**Novelty:** 4
**Technical Quality:** 5

**Review:**

By leveraging the strengths of large language models (LLMs), the authors present an efficient framework for taxonomy completion that captures both semantic and structural information in a unified manner. They further enhance the model's understanding of the taxonomy through fine-tuning with contrastive learning and mixed data augmentation.

However, the complexity of the model and the vague description make it challenging to reproduce the results. Therefore, I believe the authors need to focus more on improving the reproducibility of their work.
If the reproducibility of the model can be guaranteed, I can improve my score.

**Questions:**

1. How does the model make inferences? Need to traverse the given taxonomy graph?

**Reviewer Confidence:**

3: The reviewer is confident but not certain that the evaluation is correct

**Scope:**

4: The work is relevant to the Web and to the track, and is of broad interest to the community

---

### Official Review · Reviewer_aBEX · 2024-12-02

**Novelty:** 6
**Technical Quality:** 6

**Review:**

The paper introduces COMI, a framework for taxonomy completion that leverages large language models
to integrate semantic and structural information.
The semantic compression technique allows for the efficient encoding of long concept descriptions into single-token
representations, enabling LLMs to process taxonomic structures with reduced computational and memory costs.
To further improve the training, more examples are generated by mixup augmentation strategy,
applied at both input and manifold levels.

The paper is well-written and easy to follow. Nevertheless, some more details in the approach section would help to understand it better e.g. how the compressed version of a concept is again used in a prompt (instead of the token, the hidden layer is modified? - see also the question 1).
When reading the introduction of the paper, I thought that structural modeling would also include to strictly enforce an acylic graph
which might be an additional future work.

The train, validation, and test set is generated by randomly selecting classes from the ontology.
When doing this randomly, it is not really clear how often top-level, mid-level, or leaf concepts are used for testing.
Because a top level class might not have any other parents whereas leaf concepts do not have any children.
How does this effect the approach?

During the evaluation, the authors mentioned a significant improvement in the representation-based taxonomy completion
task. I didn't see any statistical test that underlines this
(it can be short, but if the word significant is used, it should also be mathematically argued).

The code and compressed tokens should already be available for review and not only after publication.

In conclusion, this paper makes a substantial contribution to the field of taxonomy completion,
offering a framework that is both efficient and accurate. COMI’s innovative use of semantic compression
and mixup augmentation sets a high benchmark for future research and
practical applications in knowledge structuring.

Minor:
- In Figure 1 the word "of" is in two lines

**Questions:**

1) How are the representations h_p, h_c, and h_s provided in the prompt in section 3.2.2?
2) How can the authors make sure that the same proportion of top-level, mid-level, or leaf concepts are selected?
3) Where is the code for reviewing?

**Reviewer Confidence:**

2: The reviewer is willing to defend the evaluation, but it is likely that the reviewer did not understand parts of the paper

**Scope:**

4: The work is relevant to the Web and to the track, and is of broad interest to the community

---

### Official Review · Reviewer_AL8E · 2024-12-03

**Novelty:** 4
**Technical Quality:** 5

**Review:**

An efficient taxonomy completion framework for taxonomy completion with large language models

Strengths:
1. Taxonomy completion is an important topic.
2. Semantic Compression, Contrastive Structure Modeling, and Mixup Enhanced Structure Discrimination can help improve the model performance.
3. Experiments show the model accuracy and 284x faster inference

Weaknesses:
1. A qualitative example can help demonstrate the performance.

**Questions:**

1. If fine-tuning llama3.1-8B, will the performance improved? How is the performance when using big models like GPT-4o?

**Reviewer Confidence:**

2: The reviewer is willing to defend the evaluation, but it is likely that the reviewer did not understand parts of the paper

**Scope:**

3: The work is somewhat relevant to the Web and to the track, and is of narrow interest to a sub-community

---

### Official Review · Reviewer_9K1p · 2024-12-03

**Novelty:** 4
**Technical Quality:** 4

**Review:**

The paper introduces COMI, a novel framework designed to improve taxonomy completion by leveraging LLMs to integrate semantic and structural information. COMI addresses the inherent challenge of aligning these two aspects by compressing node semantics into token representations, thus facilitating efficient LLM processing. The approach combines semantic compression with contrastive learning and mixup data augmentation to generate effective and diverse training samples. The experimental results indicate that COMI significantly speeds up inference while achieving state-of-the-art performance in taxonomy completion tasks.

Pros
The paper presents a unique approach to integrating semantic and structural information, which has traditionally been difficult to balance in taxonomy completion tasks.
The writing is clear and descriptive.

Cons
The methodology involves several complex steps and manipulations, inlcuding semantic compression and contrastive learning with mixup, which might be challenging for practical implementation without extensive computational resources.
A more in-depth evaluation of general implementation overheads would also be helpful.
While the paper is detailed, it could benefit from a clearer organization. Some sections, particularly the ablation studies and further discussions, could be more succinctly presented. Consider restructuring the narrative to improve readability.

**Questions:**

Given the complexity of the method, what are the computational resource requirements, especially in terms of memory and processing power, for training and inference?
How could this framework be adapted to continuously evolving taxonomies where new concepts are added dynamically?
Can the authors provide more insights into the types of errors the model makes? How do these errors compare across the different datasets?
How does the choice of semantic compression (one-word representation) affect the overall performance? Are there cases where such compression leads to loss of essential semantic context?
Could the mixup data augmentation strategies be extended or adapted for other types of hierarchical data or non-hierarchical structured data?

**Reviewer Confidence:**

2: The reviewer is willing to defend the evaluation, but it is likely that the reviewer did not understand parts of the paper

**Scope:**

2: The connection to the Web is incidental, e.g., use of Web data or API